# Topological-Insulator Nanocylinders

Michele Governale[1,★] and Fabio Taddei[2]

**1** School of Chemical and Physical Sciences, Victoria University of Wellington, Wellington 6140, New Zealand

**2** NEST, Istituto Nanoscienze-CNR and Scuola Normale Superiore, I-56126 Pisa, Italy

★ michele.governale@vuw.ac.nz

## Abstract

Nanostructures, such a quantum dots or nanoparticles, made of three-dimensional topological insulators (3DTIs) have been recently attracting increasing interest, especially for their optical properties. In this paper we calculate the energy spectrum, the surface states and the dipole matrix elements for optical transitions with in-plane polarization of 3DTI nanocylinders of finite height $L$ and radius $R$. We first derive an effective 2D Hamiltonian by exploiting the cylindrical symmetry of the problem. We develop two approaches: the first one is an exact numerical tight-binding model obtained by discretising the Hamiltonian. The second one, which allows us to obtain analytical results, is an approximated model based on a large-$R$ expansion and on an effective boundary condition to account for the finite height of the nanocylinder. We find that the agreement between the two models, as far as eigenenergies and eigenfunctions are concerned, is excellent for the lowest absolute value of the longitudinal component of the angular momentum. Finally, we derive analytical expressions for the dipole matrix elements by first considering the lateral surface alone and the bases alone, and then for the whole nanocylinder. In particular, we focus on the two limiting cases of tall and squat nanocylinders. The latter case is compared with the numerical results finding a good agreement.

# 1  Introduction

Over the last years the interest in topological order in condensed matter systems has exploded: many different classes of topological materials have been discovered and many possible experimental implementations have been proposed. Relevant examples are three-dimensional topological insulators (3DTIs) [1] which are materials having an insulating bulk, while exhibiting electronic conducting surface states that are protected against perturbations that do not break time-reversal symmetry (see reviews in Refs. [2–5]). Low-energy excitations, within the bulk band gap, consists of two-dimensional Dirac fermions.

In the form of nanoparticles or quantum dots (QDs), 3DTIs have been recently predicted to exhibit interesting optical properties [6–11], especially in the THz range. Possible applications include optoelectronic devices to produce THz radiation in the so-called THz gap [12,13], and optically controlled quantum elements for quantum computing [14]. An interesting optical effect, based on an electron-mediated phonon-light coupling in the THz range, has been recently measured in $Bi_2Se_3$ in Ref. [15].

The topological properties of nanoparticles, in particular related to quantum confinement, were theoretically investigated for various shapes [16–18] including disks [19] and nanospheres [7]. The fabrication, the electrical and optical measurements of such 3DTI nanoparticles or QDs have been reported by several groups over the last decade [20–32]. Interestingly, $Bi_2Se_3$ 3DTI nanoparticles also find applications in the bio-medical field, for example for the protection they provide against ionizing radiation based on their superior antioxidant activities and electrocatalytic properties [33], or for cancer therapy techniques (see Ref. [34] for a review).

Nanowires too have attracted considerable attention both theoretically and experimentally [35–54], also for the interesting quantum-confinement effects due to a finite cross section [55–58]. In particular, in Refs. [55, 56] the properties of a finite-radius 3DTI infinite cylinder were theoretically explored. In Ref. [55], an approximate analytic model supplemented by a numerical scheme was introduced to study the quantum interference effects on the low-energy spectrum of $Bi_2Se_3$ nanowires. In Ref. [56], the envelope-function description of the TI bulk band structure developed in Refs. [59,60] was used to determine the dependence of its energy

spectrum and eigenfuctions on the radius $R$. An approximate expression for the eigenenergies was obtained in the limit of large radii, up to second order in $1/R$. The analytical functional form of the eigenfunctions, which is valid irrespective of the radius of the wire, was used to calculate the dipole matrix elements for optical transitions. The case of a nanocylinder with a finite height was considered in Ref. [61] aiming at studying the spin and parity structure of the eigenstates. In particular, an approximate surface Dirac-fermion approach was used to determine analytically the energy spectrum.

In this paper we investigate finite-height 3DTI nanocylinders with the aim to single out the effects related to the finiteness of the height $L$ on the electronic energy spectrum, the wavefunctions of surface state and the optical absorption properties. Regarding the latter, in particular, two important mechanisms affecting the optical absorption of a finite cylinder are neglected in the infinite-height limit. First, the finite height changes the energies of the surface states and therefore the frequencies of the absorption lines. Second, the wavefunction on the sides of the cylinder is affected by the finite height and moreover contributions from the bases also need to be accounted for. The fact that the wavefunction is different in the case of finite-height cylinders implies that the dipole matrix elements, which determine the strength of the absorption, are different than the one computed in the infinite-height limit. We are further motivated by experiments which are currently under progress [62], where the absorption of THz radiation is measured in cylindrical QDs of various sizes.

From a more general point of view, a very interesting issue related to the finiteness of the height $L$ is the theoretical problem of finding the correct quantisation condition for the longitudinal momentum entering the wavefunction of the lateral side surface of the cylinder. Indeed, the side surface state of the cylinder, when it approaches the top/bottom bases, bends and merges with the surface states on the bases and hard-wall boundary conditions cannot be imposed.

Depending on the values of $L$ and $R$, we consider nanocylinders whose geometry ranges from the one of a squat QD (when $L \ll R$) to the one of a tall nanorod (when $L \gg R$). We will make use of two models. Both of them are based on an effective two-dimensional (2D) Hamiltonian obtained leveraging the cylindrical symmetry of the problem. More precisely, a set of independent 2D problems are obtained for each value of the quantum number $j$, which is the eigenvalue of the projection along the longitudinal direction $z$ of the electronic total angular momentum. The first model exactly describes a finite-height cylinder: it is a numerical tight-binding model where the lattice spacings for the two spatial coordinates are allowed to take different values. The second is an approximated analytical model that describes the surface states of the lateral side of the nanocylinder (in the large radius limit) by accounting for the presence of the two bases through an effective boundary condition. The latter provides the quantization rule for the momentum along $z$ through which we get approximated expressions for the eigenenergies of a nanocylinder and for the wavefunction of its lateral surface. We stress that the quantization condition we obtain differs from the one derived in Ref. [61], where a Dirac-fermion (surface) theory was employed, instead of the full (bulk) Hamiltonian of the lateral side of the cylinder we use here.

To check the quality of such approximated model we have compared its results with the ones obtained with the exact numerical model. As far as eigenenergies are concerned, the agreement is excellent for $j = \pm 1/2$ and in general yields the correct dependence on the radius R of the cylinder. The wavefunction of the lateral surface is also well reproduced by the analytical model at least not too close to the top and bottom bases. In particular, we have calculated numerically the probability density for the four different components of the

spinor wavefunction, emphasizing peculiar behaviours such as the fact that the probability density on the two bases is small in the region around their centres.

Finally, the approximated model has been used to derive analytical expressions for the dipole matrix elements for optical transitions with in-plane polarization. We calculate the contribution from the lateral side alone and from the bases alone, which allows us to derive the matrix elements in the limits of a tall rod and of a squat QD, respectively. In the latter case, we have checked the validity of the analytical expression using the numerical exact model, finding a good agreement at least for the transitions at the lowest energy.

The paper is organized as follows. In section 2 we specify the low-energy Hamiltonian of a class of 3DTI materials taking into account the cylindrical symmetry, and derive the effective 2D problem. In section 3 we show the derivation of the two models for a nanocylinder: the analytical model for the side of the cylinder, its bases and the effective boundary condition in the first three sub-sections, while the numerical approach is detailed in the forth sub-section. The results for the two models (eigenenergies, wavefunctions and optical transitions) are shown and discussed in section 4.

## 2 Hamiltonian of a 3DTI

We consider a nanocylinder, with radius $R$ and height $L$, made of a 3DTI material such as $Bi_2Se_3$. The axis of the cylinder is along the $z$ direction. As a model Hamiltonian we consider the one derived in Refs. [59, 60], namely

$$H_{3D} = \begin{pmatrix} m(\mathbf{p}) & Bp_z & 0 & Ap_- \\ Bp_z & -m(\mathbf{p}) & Ap_- & 0 \\ 0 & Ap_+ & m(\mathbf{p}) & -Bp_z \\ Ap_+ & 0 & -Bp_z & -m(\mathbf{p}) \end{pmatrix}, \tag{1}$$

which describes the low-energy properties of the bulk 3DTI. Here $\mathbf{p} = (p_x, p_y, p_z)$ is the momentum operator, $m(\mathbf{p}) = m_0 + m_1 p_z^2 + m_2(p_x^2 + p_y^2)$ is the mass term and $p_\pm = p_x \pm ip_y$. The Hamiltonian (1) is written in the basis of the four states closest to the Fermi energy at the $\Gamma$ point, i.e. $\{|P1_z^+ \uparrow\rangle, |P2_z^- \uparrow\rangle, |P1_z^+ \downarrow\rangle, |P2_z^- \downarrow\rangle\}$. The label $P1(2)_z$ indicates that they are relative to the atomic $p_z$ orbitals of the two different atoms in the material, the superscript $\pm$ refers to their parity, while $\uparrow (\downarrow)$ is the spin. The mass coefficients $m_0$, $m_1$ and $m_2$, as well as the coefficients $A$ and $B$ of the linear-momentum terms depend on the material. For example, for $Bi_2Se_3$ we have $m_0 = -0.169$ eV, $m_1 = 3.353$ eVÅ$^2$, $m_2 = 29.375$ eVÅ$^2$, $A = 2.513$ eVÅ, and $B = 1.836$ eVÅ [63]. Note that when the sign of $m_0/m_2$ is negative, the material is in the topological phase characterized by surfaces states represented by gapless Dirac cones.

Because of the cylindrical symmetry, it is useful to rewrite the effective Hamiltonian (1) in cylindrical coordinates $(r, \phi, z)$ obtaining [56, 64]

$$H_{3D} = \begin{pmatrix} m & -iB\partial_z & 0 & -Ae^{-i\phi}D_+ \\ -iB\partial_z & -m & -Ae^{-i\phi}D_+ & 0 \\ 0 & -Ae^{i\phi}D_- & m & iB\partial_z \\ -Ae^{i\phi}D_- & 0 & iB\partial_z & -m \end{pmatrix}, \tag{2}$$

where $m = m_0 - m_2(\partial_r^2 + \frac{1}{r}\partial_r + \frac{1}{r^2}\partial_\phi^2) - m_1\partial_z^2$ and $D_\pm = i\partial_r \pm \frac{1}{r}\partial_\phi$. We now make the following

Ansatz for the eigenfunctions of the Hamiltonian (2)

$$\Psi_j(r,\phi,z) = \frac{e^{ij\phi}}{\sqrt{2\pi}} \frac{1}{\sqrt{r}} \begin{pmatrix} u_1(r,z)e^{-\frac{i}{2}\phi} \\ u_2(r,z)e^{-\frac{i}{2}\phi} \\ u_3(r,z)e^{\frac{i}{2}\phi} \\ u_4(r,z)e^{\frac{i}{2}\phi} \end{pmatrix}, \tag{3}$$

where the quantum number $j$, which takes half-integer values, is the eigenvalue of the projection along $z$ of the total angular momentum $\mathbf{J}$. Inserting the Ansatz in the Schrödinger equation, i.e. $H_{3D}\Psi_j(r,\phi,z) = E\Psi_j(r,\phi,z)$, we obtain

$$\begin{pmatrix} \bar{m} + V_{c,-} - E & -iB\partial_z & 0 & -iA\left[\partial_r + \frac{j}{r}\right] \\ -iB\partial_z & -\left[\bar{m} + V_{c,-} + E\right] & -iA\left[\partial_r + \frac{j}{r}\right] & 0 \\ 0 & -iA\left[\partial_r - \frac{j}{r}\right] & \bar{m} + V_{c,+} - E & iB\partial_z \\ -iA\left[\partial_r - \frac{j}{r}\right] & 0 & iB\partial_z & -\left[\bar{m} + V_{c,+} + E\right] \end{pmatrix} \begin{pmatrix} u_1 \\ u_2 \\ u_3 \\ u_4 \end{pmatrix} = 0, \tag{4}$$

where we have introduced the following definitions

$$\bar{m} = m_0 - m_2\partial_r^2 - m_1\partial_z^2 \tag{5}$$

$$V_{c,\pm} = \frac{m_2}{r^2}\left(j^2 \pm j\right). \tag{6}$$

## 2.1 Effective 2D Hamiltonian

Interestingly, Eq. (4) suggests that for each value of the quantum number $j$ we can define an effective two-dimensional problem with an effective Hamiltonian given by

$$H_{\mathrm{eff},j} = \begin{pmatrix} \bar{m} + V_{c,-} & -iB\partial_z & 0 & -iA\left[\partial_r + \frac{j}{r}\right] \\ -iB\partial_z & -\bar{m} - V_{c,-} & -iA\left[\partial_r + \frac{j}{r}\right] & 0 \\ 0 & -iA\left[\partial_r - \frac{j}{r}\right] & \bar{m} + V_{c,+} & iB\partial_z \\ -iA\left[\partial_r - \frac{j}{r}\right] & 0 & iB\partial_z & -\bar{m} - V_{c,+} \end{pmatrix} \tag{7}$$

and acting on the spinor wavefunction $\Phi_j(r,z) = (u_{1,j}(r,z), u_{2,j}(r,z), u_{3,j}(r,z), u_{4,j}(r,z))^{\mathrm{T}}$. Due to the presence of the factor $1/\sqrt{r}$ in the Ansatz (3), the normalisation condition takes the standard form for a wavefunction in 2D Cartesian coordinates, that is

$$\int_0^L dz \int_0^R dr\, \Phi_j(r,z)^\dagger \Phi_j(r,z) = 1. \tag{8}$$

# 3 Models of a finite-size cylinder

The surface states for a finite-size cylinder of TI are localised around the *entire* surface, that is both the bases and the lateral side. For a cylinder of height $L$ and radius $R$, the wavefunction

$\Phi_j$ satisfies the boundary conditions

$$\Phi_j(r = 0, z) = 0 \quad \forall \ z \tag{9a}$$

$$\Phi_j(r = R, z) = 0 \quad \forall \ z \tag{9b}$$

$$\Phi_j(r, z = 0) = 0 \quad \forall \ r \tag{9c}$$

$$\Phi_j(r, z = L) = 0 \quad \forall \ r, \tag{9d}$$

and the normalisation condition Eq. (8). The effective 2D problem corresponds to confinement in a two-dimensional box.

As discussed in Sec. 3.4, for each value of $j$, the full surface state for arbitrary values of $R$ and $L$ can be numerically calculated by discretising the Hamiltonian (7) on a finite grid thus obtaining a 2D-tight binding model. Analytically, however, the full surface state can be obtained through an approximate method, based on a multiple-scattering argument, that matches the surface states on the side of the cylinder with those on the two bases (see Sec. 3.3). To do so, we first we need to find the surface states for the side and the bases.

## 3.1 Side: Large-$R$ expansion

We start with the side of the cylinder and we perform a large-$R$ expansion following a different approach with respect to the one of Ref. [56, 64]. We consider the effective 2D Hamiltonian (7) and write it as

$$H_{\text{eff},j} = H_0 + H_{1,j}, \tag{10}$$

where $H_0$ describes in-plane (transverse section) motion in leading order in $1/R$

$$H_0 = \begin{pmatrix} m_0 - m_2\partial_r^2 & 0 & 0 & -iA\partial_r \\ 0 & -\left(m_0 - m_2\partial_r^2\right) & -iA\partial_r & 0 \\ 0 & -iA\partial_r & m_0 - m_2\partial_r^2 & 0 \\ -iA\partial_r- & 0 & 0 & -\left(m_0 - m_2\partial_r^2\right) \end{pmatrix}, \tag{11}$$

and $H_{1,j}$ contains corrections due to finite-$R$ and $p_z$

$$H_{1,j} = \begin{pmatrix} V_{c,-} & -iB\partial_z & 0 & -iA\frac{j}{r} \\ -iB\partial_z & -V_{c,-} & -iA\frac{j}{r} & 0 \\ 0 & iA\frac{j}{r} & V_{c,+} & iB\partial_z \\ iA\frac{j}{r} & 0 & iB\partial_z & -V_{c,+} \end{pmatrix}. \tag{12}$$

We start by calculating the eigenfunctions of $H_0$ that correspond to the surface states. We assume that the radial dependence of the wave function of the effective 2D model is of the form $e^{\lambda(r-R)}$, since we expect the surface states to be exponentially localised at the surface $r = R$. The corresponding eigenenergy is $E = 0$. As detailed in App. A, we find that the

eigenfunctions of (11) which fulfil the boundary conditions at $r = R$ are

$$\Phi_{1,k_z,j}(r,z) = \frac{1}{\sqrt{2}} \begin{pmatrix} 1 \\ 0 \\ 0 \\ i \end{pmatrix} \frac{e^{ik_z z}}{\sqrt{L}} \rho(r) \tag{13}$$

$$\Phi_{2,k_z,j}(r,z) = \frac{1}{\sqrt{2}} \begin{pmatrix} 0 \\ 1 \\ -i \\ 0 \end{pmatrix} \frac{e^{ik_z z}}{\sqrt{L}} \rho(r), \tag{14}$$

where the radial part of the wavefunction is defined as

$$\rho(r) = \frac{1}{N_{\text{sur}}} \left[ e^{\lambda_+ (r-R)} - e^{\lambda_- (r-R)} \right], \tag{15}$$

with $N_{\text{sur}}$ being a normalisation factor and

$$\lambda_{\pm} = \frac{A \pm \sqrt{A^2 + 4m_0 m_2}}{2m_2}. \tag{16}$$

The perturbation Hamiltonian $H_{1,j}$ in the space spanned by $|\Phi_{1,k_z,j}\rangle$ and $|\Phi_{2,k_z,j}\rangle$ can be written as

$$H_{1,j} \doteq \sqrt{(Bk_z)^2 + j^2 \frac{A^2}{R^2} \left( 1 - \frac{1}{2} \frac{A}{m_0 R} \right)^2} \begin{pmatrix} \cos(\theta) & \sin(\theta) \\ \sin(\theta) & -\cos(\theta) \end{pmatrix}, \tag{17}$$

with

$$\cos(\theta) = \frac{\left( \frac{A}{R} - \frac{A^2}{2m_0} \frac{1}{R^2} \right) j}{\sqrt{(Bk_z)^2 + j^2 \frac{A^2}{R^2} \left( 1 - \frac{1}{2} \frac{A}{m_0 R} \right)^2}} \tag{18}$$

and

$$\sin(\theta) = \frac{Bk_z}{\sqrt{(Bk_z)^2 + j^2 \frac{A^2}{R^2} \left( 1 - \frac{1}{2} \frac{A}{m_0 R} \right)^2}}. \tag{19}$$

Note that matrix elements between different values of $j$ are zero, because of the cylindrical symmetry of the problem. Similarly $k_z$ is also a good quantum number due to translational symmetry along $z$. More details of this calculation are provided in App. B.

The eigenergies are given by

$$E_{\pm,k_z,j} = \pm \sqrt{(Bk_z)^2 + j^2 \frac{A^2}{R^2} \left( 1 - \frac{1}{2} \frac{A}{m_0 R} \right)^2}. \tag{20}$$

We emphasise that for $k_z = 0$, Eq. (20) reproduces the result, obtained by a completely different route, of Eq. (18) in Ref. [56]. The eigenkets corresponding to the eigenenergies $E_{\pm,k_z,j}$ are

$$|\Phi_{+,k_z,j}\rangle = \cos(\theta/2)|\Phi_{1,k_z,j}\rangle + \sin(\theta/2)|\Phi_{2,k_z,j}\rangle \tag{21a}$$

$$|\Phi_{-,k_z,j}\rangle = -\sin(\theta/2)|\Phi_{1,k_z,j}\rangle + \cos(\theta/2)|\Phi_{2,k_z,j}\rangle. \tag{21b}$$

In order to be consistent with the phase of the states, we take $\theta \in [0, 2\pi]$. Finally, we note that the eigenfunctions (3) for the surface states of the original 3D problem, Eq. (2), can be written as

$$\Psi_{+,k_z,j}(r, \phi, z) = \frac{1}{\sqrt{2}} \begin{pmatrix} \cos(\theta/2)e^{-i\frac{\phi}{2}} \\ \sin(\theta/2)e^{-i\frac{\phi}{2}} \\ -i\sin(\theta/2)e^{i\frac{\phi}{2}} \\ i\cos(\theta/2)e^{i\frac{\phi}{2}} \end{pmatrix} \frac{e^{ik_z z}}{\sqrt{L}} \frac{e^{ij\phi}}{\sqrt{2\pi}} \frac{1}{\sqrt{r}} \rho(r) \tag{22a}$$

$$\Psi_{-,k_z,j}(r, \phi, z) = \frac{1}{\sqrt{2}} \begin{pmatrix} -\sin(\theta/2)e^{-i\frac{\phi}{2}} \\ \cos(\theta/2)e^{-i\frac{\phi}{2}} \\ -i\cos(\theta/2)e^{i\frac{\phi}{2}} \\ -i\sin(\theta/2)e^{i\frac{\phi}{2}} \end{pmatrix} \frac{e^{ik_z z}}{\sqrt{L}} \frac{e^{ij\phi}}{\sqrt{2\pi}} \frac{1}{\sqrt{r}} \rho(r). \tag{22b}$$

### 3.2    Top and bottom bases

In this section we write the states of the top/bottom surfaces in the absence of radial confinement. We follow Ref. [65]. However, we use polar coordinate to emphasize that the states are eigenstates of $\mathbf{J}_z$ due to the cylindrical symmetry. We assume that $L$ is large enough so that the effect of the hybridisation between the states localised at the top base and the bottom base is negligible. Therefore, we consider the two bases as independent. In the following, we provide results for the the states localised around the top base, at $z = L$. The details of this calculation are shown in App. C. The dispersion relations for the surface states of the top base are

$$E_\pm = \pm A k_\parallel, \tag{23}$$

with $k_\parallel \geq 0$. The eigenfunctions corresponding to these energies are

$$\Psi_{\text{Top},+}(r, \phi, z) = \frac{e^{ij\phi}}{\sqrt{2\pi}} \begin{pmatrix} -iJ_{j-\frac{1}{2}}(k_\parallel r)e^{-i\frac{\phi}{2}} \\ J_{j-\frac{1}{2}}(k_\parallel r)e^{-i\frac{\phi}{2}} \\ iJ_{j+\frac{1}{2}}(k_\parallel r)e^{i\frac{\phi}{2}} \\ J_{j+\frac{1}{2}}(k_\parallel r)e^{i\frac{\phi}{2}} \end{pmatrix} \frac{1}{N_{\text{base},j}} \left[ e^{\gamma_+(z-L)} - e^{\gamma_-(z-L)} \right] \tag{24a}$$

$$\Psi_{\text{Top},-}(r, \phi, z) = \frac{e^{ij\phi}}{\sqrt{2\pi}} \begin{pmatrix} iJ_{j-\frac{1}{2}}(k_\parallel r)e^{-i\frac{\phi}{2}} \\ -J_{j-\frac{1}{2}}(k_\parallel r)e^{-i\frac{\phi}{2}} \\ iJ_{j+\frac{1}{2}}(k_\parallel r)e^{i\frac{\phi}{2}} \\ J_{j+\frac{1}{2}}(k_\parallel r)e^{i\frac{\phi}{2}} \end{pmatrix} \frac{1}{N_{\text{base},j}} \left[ e^{\gamma_+(z-L)} - e^{\gamma_-(z-L)} \right], \tag{24b}$$

where $N_{\text{base},j}$ is a normalisation factor which depends on $j$ and $k_\parallel$, while the decay constants are given by

$$\gamma_\pm = \left| \frac{B \pm \sqrt{B^2 + 4m_1(m_0 + m_2 k_\parallel^2)}}{2m_1} \right|. \tag{25}$$

Analogously, the eigenfunctions corresponding to the energies $E_\pm = \pm A k_\parallel$ for the bottom base take the same form as the ones for the top base by replacing $L$ with 0.

### 3.3 Effective boundary condition

In this section we present our approximated method to obtain the eigenergies of the nanocylinders and an approximation for the wavefunction on the side surface of the cylinder. The energy for the surface state of the side of the cylinder is given by

$$E_{\pm,k_z,j} = \pm\sqrt{(Bk_z)^2 + j^2\frac{A^2}{R^2}}, \tag{26}$$

where here we neglect higher order $1/R$ corrections [see Eq. (20)]. Now, the issue with this expression is that it is not clear how to quantise the longitudinal wavevector $k_z$. An obvious choice is to use periodic boundary conditions, which would yield $k_z = n2\pi/L$. This type of boundary conditions corresponds to the cylinder being closed in a doughnut (torus) which is clearly a very different geometry than the nanocylinder. Another possibility would be to try to impose hard-wall boundary conditions at $z = 0$ and $z = L$. This turns out to be impossible as a consequence of the fact that the surface state extends over the entire surface of the nanocylinder. This means that the side state of the cylinder when it approaches the top/bottom bases bends and merges with the surface states on the bases. The solution to this problem that we pursue here is to take into account of the bases as an effective boundary condition for the surface states on the cylinder. For energies $E > 0$, we assume that the state $|\Psi_{+,k_z,j}\rangle$, in Eq. (22), gets scattered into the state $|\Psi_{+,-k_z,j}\rangle$ when it reaches the top surface, and picks up a phase $\exp(ik_\parallel 2R)$ from the propagation across the top surface and an additional negative sign for $j < 0$, due to the angular momentum. Similarly, the state $|\Psi_{+,-k_z,j}\rangle$ gets reflected at the bottom surface into $|\Psi_{+,k_z,j}\rangle$ with the same scattering phase $\exp(ik_\parallel 2R)\text{sign}(j)$. While so far we have considered the case $E > 0$, the very same argument can be carried out for $E < 0$ by changing all the state kets into $|\Psi_{-,\pm k_z,j}\rangle$. We now require that the state remains the same after a full orbit (two scattering events one at the top surface and one at the bottom one) and this yields the quantisation condition:

$$2Lk_z + 4Rk_\parallel = \tilde{n}2\pi, \tag{27}$$

where $\tilde{n}$ is an integer and $k_z > 0$. As we will see below, this latter condition yields the smaller integer value, $\tilde{n}_{\min,j}$, that $\tilde{n}$ can take for a given value of $j$. The energy dispersions for the top and bottom surfaces are $E_\pm = \pm Ak_\parallel$, while for the side surface are given in Eq. (26). Since both the side state and the bases state correspond to the same energy we have

$$k_\parallel = \sqrt{\left(\frac{B}{A}k_z\right)^2 + \left(\frac{j}{R}\right)^2}. \tag{28}$$

Finally the quantisation condition for $k_z$ reads

$$k_z + \gamma\sqrt{k_z^2 + \left(\frac{2j}{\gamma L}\right)^2} = \tilde{n}\frac{\pi}{L}, \tag{29}$$

where we have defined the parameter $\gamma = \frac{2BR}{AL}$ which is proportional to the aspect ratio $2R/L$ and the asymmetry $B/A$ of the coupling constants.

Solving the equation above for $k_z$ and taking the positive solution yields

$$k_z(\tilde{n}, j) = \frac{-\tilde{n}\frac{\pi}{L} + \sqrt{\gamma^2\left[\tilde{n}\frac{\pi}{L}\right]^2 - \left(\frac{2j}{L}\right)^2(\gamma^2 - 1)}}{\gamma^2 - 1}, \tag{30}$$

where

$$\tilde{n} \geq \text{Ceiling}[2|j|/\pi] \equiv \tilde{n}_{\min,j} \tag{31}$$

and we have explicitly indicated that $k_z$ depends on $\tilde{n}$ and $j$. We now define the quantum number $n$ such that $\tilde{n} = n + \tilde{n}_{\min,j} - 1$. After this substitution the smallest positive value of $k_z$ is obtained for $n = 1$. The quantised values for $k_z$ read

$$k_z(n,j) = \frac{-(n + \tilde{n}_{\min,j} - 1)\frac{\pi}{L} + \sqrt{\gamma^2 \left[(n + \tilde{n}_{\min,j} - 1)\frac{\pi}{L}\right]^2 - \left(\frac{2j}{L}\right)^2 (\gamma^2 - 1)}}{\gamma^2 - 1}, \tag{32}$$

with $n = 1, 2, \ldots$ This is one of the central results of this paper and a few comments are in order. First we note that the quantised values of $k_z$ depend on the angular-momentum quantum number $j$. Second, if we consider tall cylinders, that is $\gamma \to 0$, we get

$$k_z(n,j) \approx (n + \tilde{n}_{\min,j} - 1)\frac{\pi}{L} - 2\frac{|j|}{L},$$

which shows a linear dependence on the angular-momentum quantum number $j$. Thirdly, if $n \gg |j|$, we obtain for the quantised wavevectors $k_z \approx (n + \tilde{n}_{\min,j} - 1)\frac{\pi}{(1+\gamma)L}$, which resembles the results for a particle in a box.

We are now in a position to write the quantised energies for the surface state of the entire nanocylinder simply by plugging in the quantised values for $k_z$ in Eq. (26). We also change notation and we label the negative energies with negative values of $n$. In the new notation, the eigenenergies are

$$E_{n,j} = \text{sign}(n)\frac{B}{L}\sqrt{k_z(|n|,j)^2 L^2 + \frac{1}{\gamma^2}(2j)^2}, \tag{33}$$

where $k_z(|n|,j)$ is given by Eq. (32). This is another key result of this paper and it is worth examining some limiting cases. In the tall-cylinder case, $\gamma \ll 1$, the eigenenergies read

$$E_{n,j} = \text{sign}(n)\frac{A|j|}{R}\left[1 + \frac{\gamma^2}{2}\left(\frac{(|n| + \tilde{n}_{\min,j} - 1)\pi}{2|j|} - 1\right)^2\right]. \tag{34}$$

In the limit of large-quantum number $|n|$, that is $|n| \gg |j|$, we get

$$E_{n,j} \approx \text{sign}(n)B\frac{(|n| + \tilde{n}_{\min,j} - 1)\pi}{(\gamma + 1)L}, \tag{35}$$

which shows a linear dispersion in the quantised $k_z$ as expected for gapless Dirac's cone excitations. The eigenenergies determine the position of the absorption lines when we consider absorption of electro-magnetic radiation by the nanocylinders. In the next section we will benchmark the quality of the result in Eq. (33) against the full numerics.

Finally, we approximate the states on the side-surface of the nanocylinder as

$$\Psi_{n,j}(r,\phi,z) \approx \frac{e^{-ik_z L}}{\sqrt{2}}\Psi_{\pm,k_z,j}(r,\phi,z) + \text{sign}(j)e^{ik_\| 2R}\frac{e^{ik_z L}}{\sqrt{2}}\Psi_{\pm,-k_z,j}(r,\phi,z), \tag{36}$$

where the upper(lower) sign is for $n > 0(n < 0)$, $k_\| = k_\|(|n|,j) = \frac{E_{|n|,j}}{A}$ and $k_z = k_z(|n|,j)$. The knowledge of the states allows for the calculation of expectation values of observables.

We conclude by noting that the quantisation condition (32) differs from the standard quantisation condition $k_z = \frac{n\pi}{L}$.

## 3.4 Numerical approach

The effective 2D model in Sec. 2.1 can be discretised on a 2D grid whose sites are defined by $r_p = (p-1) a_r$ and $z_q = (q-1) a_z$, where $a_r$ and $a_z$ are the lattice constants along $r$ and $z$, respectively. The indices $p$ and $q$ run as follows: $p = 1, ..., N_r$ and $q = 1, ..., N_z$, with $N_r$ ($N_z$) being the number of sites in the $r$ ($z$) direction. By approximating the derivatives of the wavefunction with finite differences, we can solve the Schrödinger equation for the Hamiltonian (7), for each value of $j$. This yields a tight-binding model for the cylinder. We find the eigenfuntions $\Phi_{n,j}(r_p, z_q)$ defined on the grid and the corresponding eigenenergies $E_{n,j}$, where the quantum number $n$ is an integer. We order the energies so that $E_{n,j} < 0$ if $n < 0$ and $E_{n',j} \geq E_{n,j}$ if $n' > n$. Due to the symmetry of the Hamiltonian the eigenenergies fullfil the relation $E_{-n,j} = -E_{n,j}$. Finally, for the discretised wavefunction the normalisation reads $\sum_{p,q} \Phi_{n,j}(r_p, z_q)^\dagger \Phi_{n,j}(r_p, z_q) = 1$.

## 4 Results

Here we use the two models developed in Sec. 3 to explicitly calculate the eigenenergies, eigenfunctions and dipole matrix elements for 3DTI nanocylinders. In particular, in section 4.1 we compare the results of the two models focusing on the dependence of the energy spectrum on the radius and the height of the cylinder. Moreover, we show the details of the probability density obtained using the numerical model and we compare, component by component, the wavefunctions resulting from the two models. In section 4.2 we calculate the dipole matrix elements with both the numerical and the analytical models for optical transitions and show their dependence on the radius of the cylinder. We express the energy in units of $|m_0|$ and all lengths in units of $R_0 = |A/m_0|$.

### 4.1 Eigenenergies and Eigenfunctions

In Fig. 1, we show the quantised energy levels (with $n = 1, 2, 3$) of the full surface states as functions of the radius $R$, for fixed $L = 10R_0$, and for different values of $j$ in the different panels [$j = 1/2$ in panel (a), $j = 3/2$ in panel (b), $j = 5/2$ in panel (c), and $j = 7/2$ in panel (d)]. The results of the exact numerical approach are represented by filled circles, while the results of the approximated analytical model are represented by solid lines. We notice that the analytical model agrees exceptionally well with the numerical calculation for the lowest value of $j$, panel (a). For $j = 3/2$ and $5/2$, panels (b) and (c), the approximated model still captures the overall behaviour of the energy spectrum, but its accuracy degrades. Interestingly, the accuracy of the model improves again for $j = 7/2$. Moreover, as expected, the analytical model performs better for larger values of $R$ as it is based on a large-$R$ approximation.

In Fig. 2, we show the dependence on the height $L$ of the eigenenergies of the full surface states, in analogy to what was done in Fig. 1. We notice that the dependence on $L$ is much weaker than the dependence on the radius $R$. Also in Fig. 2, the analytical calculation is particularly accurate for $j = 1/2$ (apart from the case $L = 2.5R_0$, which however corresponds to a very squat cylindrical QD), with the accuracy becoming worst for $j = 3/2$. It is noteworthy that such a simple model does so well in describing the topological states of a finite-size cylinder of TI.

Let us now consider the eigenfunctions $\Phi_{n,j}(r, z)$, introduced in sec. 3.4, and define the

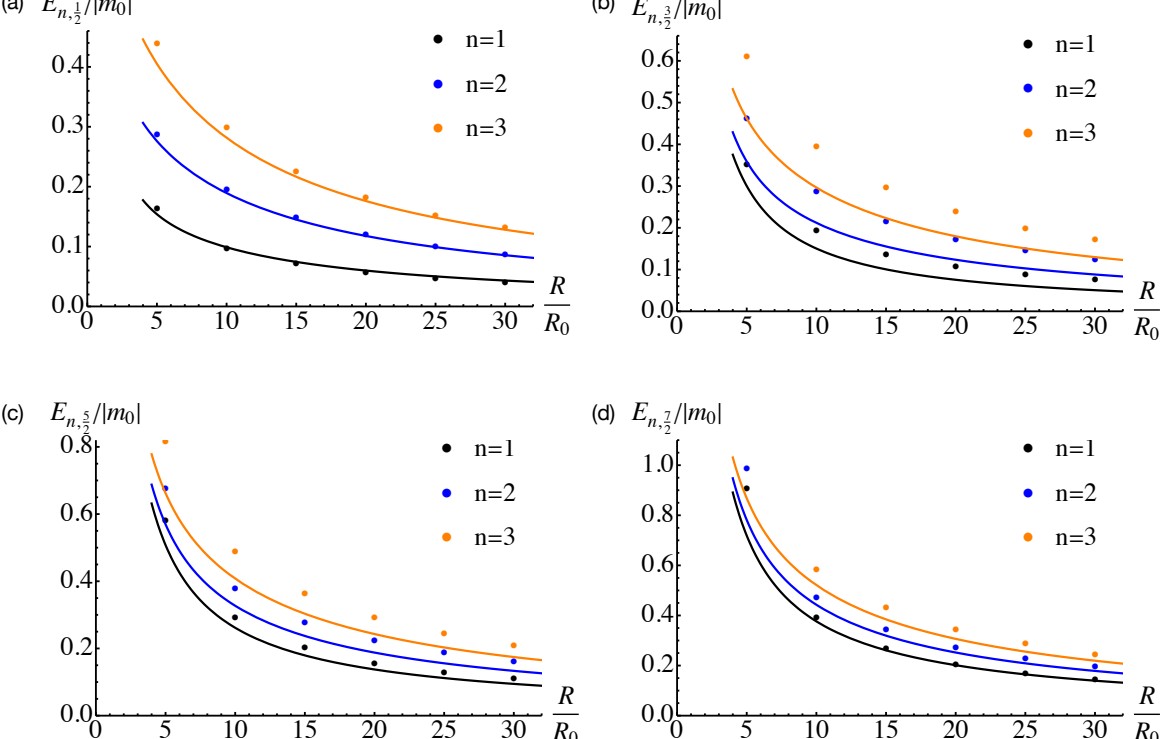

Figure 1: The quantised energy in units of $|m_0|$ of the surface states for a nanocylinder of height $L = 10R_0$ as a function of the radius $R$, for different values of the quantum number $n$ and $j = 1/2$ panel (a), $j = 3/2$ panel (b), $j = 5/2$ panel (c), and $j = 7/2$ panel (d). The filled circles correspond to the results of the numerical calculation with discretisation constants $a_r = 0.25R_0$ and $a_z = 0.05R_0$, while the solid lines are obtained with the analytical model.

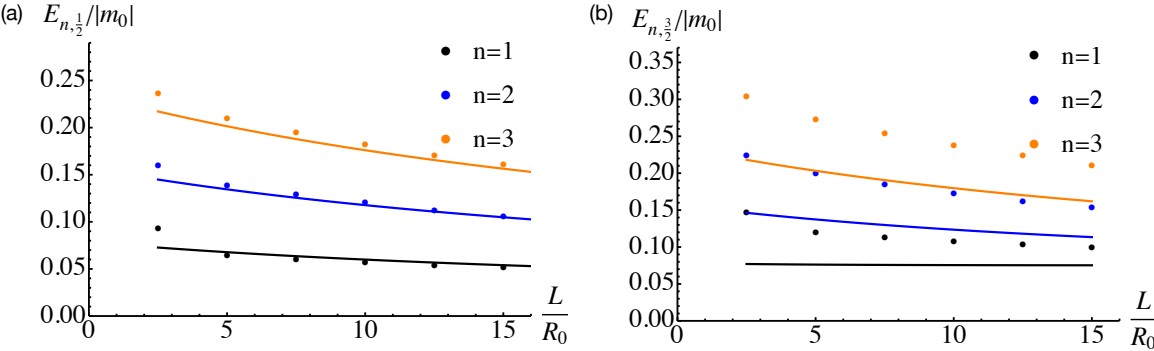

Figure 2: The quantised energy in units of $|m_0|$ of the surface states for a nanocylinder of radius $R = 20R_0$ as a function of the height $L$, for different values of the quantum number $n$ and $j = 1/2$ panel (a), and $j = 3/2$ panel (b). The filled circles correspond to the results of the numerical calculation with discretisation constants $a_r = 0.25R_0$ and $a_z = 0.05R_0$ while the solid lines are obtained with the analytical model.

probability density as $\rho_{n,j}(r,z) = \Phi_{n,j}^\dagger(r,z)\Phi_{n,j}(r,z)$. In Fig. 3 we plot $\rho_{n,j}(r,z)$, scaled so that its maximum value is 1, with $n = 1$ and $j = 1/2$ (this corresponds to the state having the lowest positive value of the energy) for a nanocylinder with radius $R = 30R_0$ and height $L = 10R_0$. As expected, the surface state extends over the entire surface of the nanocylinder. The density vanishes for $r = R$, for $z = 0$ and for $z = L$, as required by the hard-wall boundary condition. Interestingly, the density on the two bases appears to be small in the region around their centres ($r = 0$).

In Fig. 4 we plot the cuts along the dashed lines indicated in the density plot in Fig. 3, emphasising the contribution of each component of the eigenfunction (see Sec. 2.1). More precisely, we use the notation $\rho_{n,j}^{(i)}(r,z)$, with $i = 1, ..., 4$, to denote the four different components of the overall probability density. Panel (a) shows the probability density as a function of $r$ along the horizontal dashed line in Fig. 3, which corresponds to a value of $z$ close to the peak near the top surface, whose expression is $z = z_u = L - R_0/4$. Notice that the components 1 and 2 are nearly equal and are non-zero everywhere on the top base, while decreasing to zero approaching its centre ($r = 0$). The components 3 and 4 are also nearly equal, but are negligibly small over a large portion of the top base around its centre, roughly up to $r = R/2$. The overall probability density increases steadily with the radial coordinate from zero at $r = 0$ and then drops back to zero at $r = R$. Panel (b) shows the probability density as a function of $z$ at $r = \bar{r} = R - R_0 \left[\frac{1}{2} + \frac{m_2}{A^2}|m_0|\right]$, which corresponds to the vertical dashed line in Fig. 3 that is located approximatively at the position of the density peak on the side of the cylinder (away from the bases). Here, components 1 and 4, as well as components 2 and 3, are nearly equal apart from the boundary regions. Interestingly, $\rho_{n,j}^{(2)}$ and $\rho_{n,j}^{(3)}$ are vanishing over almost the whole height of the cylinder. The total probability density exhibits two peaks very close to $z = 0$ and $z = L$ and it is constant in between.

We conclude the section by showing the quality of the predictions provided by the analytical model by comparing the wavefunctions obtained with it to the ones calculated by means of the tight-binding model. We consider a 3DTI nanocylinder with $R = 30R_0$ and $L = 10R_0$

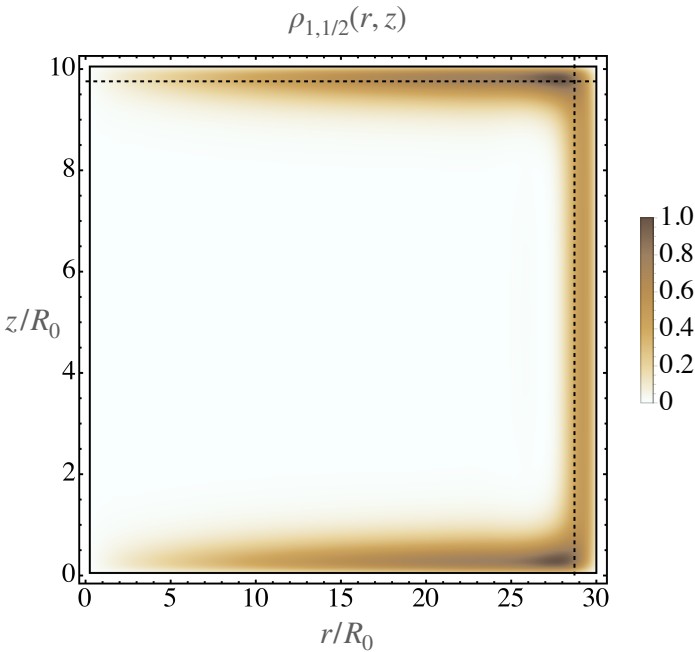

Figure 3: Probability density, scaled so that its maximum value is 1, for the state with $n = 1$ and $j = 1/2$ for $R = 30R_0$ and $L = 10R_0$, obtained by means of the numerical tight-binding model with discretisation constants $a_r = 0.25R_0$ and $a_z = 0.05R_0$. As expected, the surface state extends over the entire surface of the cylinder. The vertical dashed line is located at $r = \bar{r} = R - R_0 \left[ \frac{1}{2} + \frac{m_2}{A^2} |m_0| \right]$, which is approximatively the position of the peak of the probability density on the side of the cylinder (away from the bases). The horizontal dashed line is located at $z = z_u = L - R_0/4$ and this is close to the peak near the top surface.

and we choose the quantum numbers $n = 1$ and $j = 1/2$. The numerical results are obtained by means of the numerical tight-binding model with discretisation constants $a_r = 0.25R_0$ and $a_z = 0.05R_0$. In Fig. 5 the real (black) and imaginary (blue) parts of the wavefunctions $u_i$ (with $i = 1, ..., 4$), defined in Eq. (3), are plotted as functions of $z$. We have fixed $r = \bar{r}$, i.e. $z$ is running along the vertical dashed line in Fig. 3. The four panels are relative to the four different components of the wavefunction. Solid lines are obtained with the analytical model, while filled circles are calculated with the numerical model. On the one hand, the imaginary parts in (a) and (c), and the real parts in (b) and (d), show an excellent agreement away from the bases located at $z = 0$ and $z = L$, i.e., when $z$ is far from the bases by a few units of $R_0$. In fact, very close to such points the numerical exact results present peaks which are not captured by the analytical model. This is due to the fact that the approximated analytical model takes into account the top and bottom bases only by means of an effective boundary condition for the side-surface states and not by actually matching wavefunctions. On the other hand, in panels (a) and (c) the real part is zero, while in panels (b) and (d) is the imaginary part which vanishes: the two models agree on that, i.e. on the relative phases between the different components of the wavefunction. In conclusion, the wavefunctions obtained analytically are a very good approximation of the exact wavefunctions as long as we are a few units of $R_0$ away

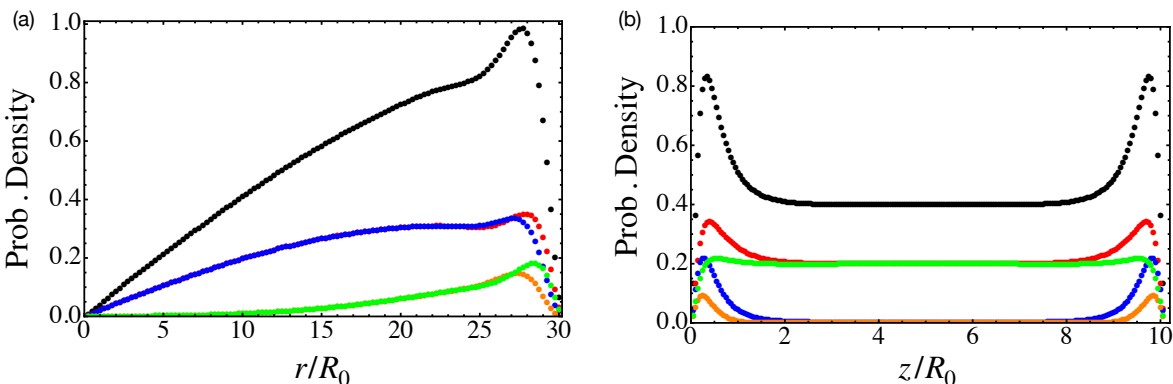

Figure 4: Cuts of the density plot of the probability density in Fig. 3. The black symbols refer to the total density $\rho_{1,1/2}$. The other colours refer to: red $\rho_{1,1/2}^{(1)}$; blue $\rho_{1,1/2}^{(2)}$; orange $\rho_{1,1/2}^{(3)}$; green $\rho_{1,1/2}^{(4)}$. Panel (a) is the cut along the horizontal dashed line at $z = z_u$, while panel (b) is the cut along the vertical dashed line located at $r = \bar{r}$.

from the bases.

## 4.2 Dipole matrix elements

Optical spectroscopy provides a useful tool to study the electronic structure in semiconducting low-dimensional systems. Optical transitions are mediated by the matrix elements of the electric-dipole operator **d** which determine the amplitudes of absorption. In the case of transitions between surface states of topological-insulator cylinders of realistic radii the frequencies of the absorption lines are in the THz range. We refer to Ref. [56] for the derivation of the operator **d**. Here we use the wavefunctions calculated in the previous sections to evaluate the dipole matrix elements for optical transitions with in-plane polarisation of 3DTI nanocylinders. As a starting point we derive the analytical expressions of the matrix elements accounting for the side surface only, and for the bases only. Using the approximated model we then obtain the analytical expression for the matrix elements of a finite-size nanocylinder. From these results we can address the limiting cases of a tall nanorod ($\gamma \ll 1$) and of a squat cylindrical QD ($\gamma \gg 1$). We can do that by using the quantisation rule for $k_z$, Eq. (32), obtained by imposing the boundary condition in Sec. 3.3. The numerical model is finally used for calculating the matrix elements of a squat cylindrical QD, finding a good agreement with the analytical result at least for the lowest transition energies.

We start considering the lateral side surface states alone of a nanocylinder of height $L$. The dipole matrix elements, which determine the radiation absorption amplitude, for circular polarisation between the states given in Eqs. (22) can be calculated easily and are given by

$$\frac{\langle \Psi_{+,k_z',j'} | d_x \pm i d_y | \Psi_{-,k_z,j} \rangle}{e R_0} = \delta_{j',j\pm 1} F_{k_z',k_z}(L) \sin\left( \frac{\theta_{j\pm 1,k_z'} - \theta_{j,k_z}}{2} \right) \left( \frac{R}{R_0} + s - \frac{m_2}{A R_0} \right), \quad (37)$$

where $s = \text{sign}(A/m_0) < 0$ in the topological state and $F_{k_z',k_z}(L) = \frac{1}{L} \int_0^L dz \, \exp[i(k_z - k_z')z]$. The mixing angles $\theta_{j,k_z}$ are given by Eqs. (18) and (19). For a very large value of $L$ this

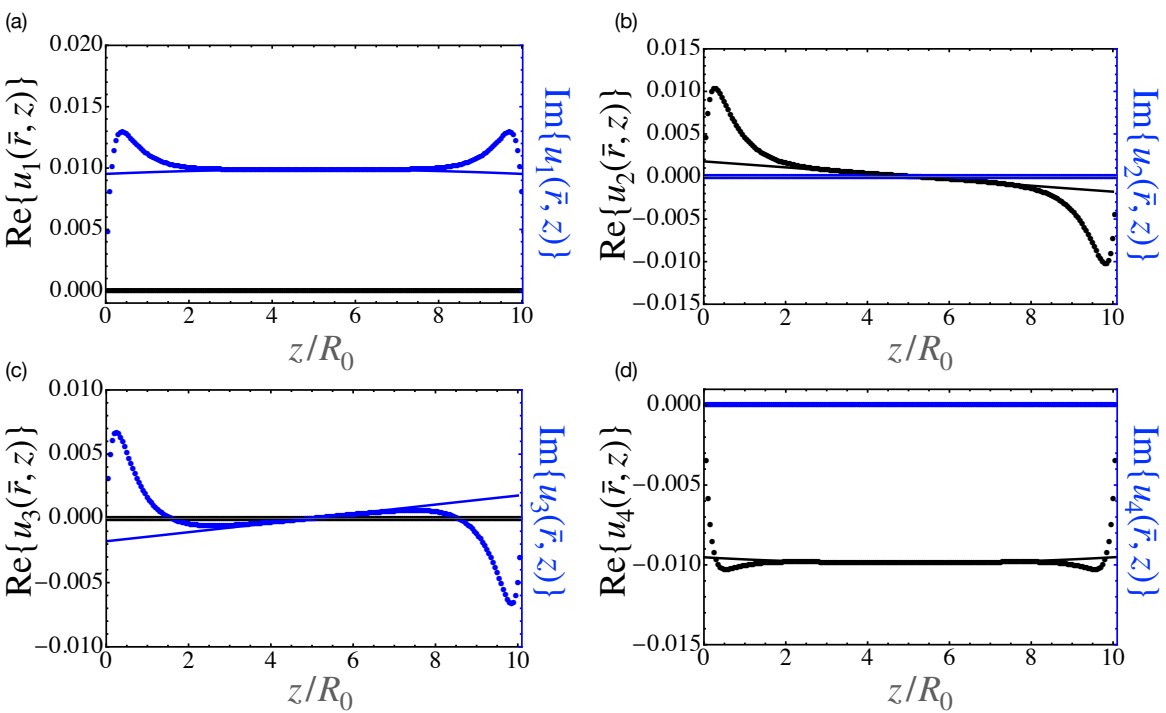

Figure 5: The four components of the wavefunctions $u_i$ (with $i = 1, ..., 4$), defined in Eq. (3), are plotted as functions of $z$ for the quantum numbers $n = 1$ and $j = 1/2$. The four panels (a), ..., (d) refer to the components 1, ...,4, respectively. They are calculated for a nanocylinder with $R = 30R_0$ and $L = 10R_0$ at a fixed $r = \bar{r}$, which approximatively corresponds to the position of the peak of the probability density on the side of the cylinder. Solid lines are obtained through the approximated analytical model, while the filled circles are the result of the exact numerical model obtained by means of the numerical tight-binding model with discretisation constants $a_r = 0.25R_0$ and $a_z = 0.05R_0$

expression approximates the matrix elements for the case of an infinite 3DTI nanowire. As an example, we look at the transition $|\Psi_{-,k_z,-1/2}\rangle \rightarrow |\Psi_{+,k_z,1/2}\rangle$. For the smallest value allowed for $k_z$, this transition happens at the lowest frequency for which we have absorption. It is also the one with the largest probability. The matrix element for this transition reads

$$\frac{\langle \Psi_{+,k_z,\frac{1}{2}} | d_x + i d_y | \Psi_{-,k_z,-\frac{1}{2}} \rangle}{e R_0} = -\text{sign}(k_z) \frac{A}{2R} \frac{1}{E_{+,1/2,k_z}} \left[ \frac{R}{R_0} - 1 - \frac{m_2}{A R_0} \right], \qquad (38)$$

where we have taken $s = -1$ and we have taken the mixing angles without the $1/R^2$ corrections.

We now look at the dipole matrix elements between the surface states of the top/bottom bases given by Eqs. (24). In the calculation of the matrix elements we integrate over the area of a disk of radius $R$. In particular we compute the matrix element for $d_x \pm i d_y$ between the

states with $j = \pm 1/2$ and opposite energies and we find

$$\frac{\langle \Psi_{\text{Top},+,k_\parallel,\pm 1/2}|d_x \pm id_y|\Psi_{\text{Top},-,k_\parallel,\mp 1/2}\rangle}{eR_0} = \frac{R/R_0}{k_\parallel R \left[\frac{J_0(k_\parallel R)}{J_1(k_\parallel R)}\right]^2 + k_\parallel R - \frac{J_0(k_\parallel R)}{J_1(k_\parallel R)}}. \qquad (39)$$

Let us now consider a 3DTI nanocylinder with radius $R$ and height $L$. Using the approximate analytical state of Eq. (36) we can calculate (see App. D) the dipole matrix element only in the limit where the contribution of the bases are negligible, that is for the case of a tall nanorod $\gamma \ll 1$. In this limit and in zeroth order in $\gamma$, we consider the absorption matrix element for $d_x + id_y$ for circular polarisation for the transition $|\Psi_{-n,-1/2}\rangle \rightarrow |\Psi_{n,1/2}\rangle$. For $k_z > 0$, we find the following expression

$$\frac{\langle \Psi_{n,1/2}|d_x + id_y|\Psi_{-n,-1/2}\rangle}{eR_0} = -\left(\frac{R}{R_0} - 1 - \frac{m_2}{AR_0}\right)\left[1 + \frac{\sin(1)}{|n|\pi - 1}\right],$$

where $n > 0$.

On the other hand, the matrix elements for the case of a squat nanocylinder is given by Eq. (39), which only accounts for the wavefunctions of the bases, and taking for $k_\parallel$ the value resulting from the boundary condition (27). We test this result by comparing it with the outcome obtained from the exact numerical model. In Fig. 6, we plot the absolute value of the matrix element of $d_x + id_y$, calculated using both models, as a function of the radius $R$ for $L = 5R_0$. The data are relative to the transitions between the states $|\Psi_{-n,-1/2}\rangle$ and $|\Psi_{n,1/2}\rangle$ for three values of $n$. Filled circles refer to the numerical model and solid lines to the analytical one. The comparison shows an excellent agreement for $n = 1$, which corresponds to the lowest energy transitions, for all values of $R$. This means that the contribution of the side surface states are unimportant. For the case $n = 2$ ($n = 3$), however, the analytical model overestimates (underestimates) the actual result, although the predicted slope is correct. The numerical curves show that the magnitude of the matrix elements are monotonically increasing with $R$, for all values of $n$ considered, and decreases as $n$ increases.

For the case of a tall nanorod we could not calculate numerically the matrix elements because for large values of $L$ the tight-binding method becomes impractical in terms of both required computational time and memory.

Finally, we show an example of absorption spectrum that can be obtained straightforwardly from the dipole matrix elements. We consider the case of circularly polarised light and we employ the tight-binding model for the nanocylinder to evaluate the matrix elements of $d_x + id_y$. We assume the low-temperature limit and the Fermi energy to be in the middle of the gap so that all the levels with energy less than zero ($n < 0$) are occupied and those with energy larger than zero ($n > 0$) are empty. In Fig. 7 we show the spectrum for a nanocylinder of radius $20R_0$ and $L = 10R_0$. The different transitions have been modelled by Gaussian curves with standard deviation $10^{-3}|m_0|$. The broadening can be due to the fact that often measurements are performed on an array of nanocylinders with slightly different radii. The energy scale $|m_0|$ for $Bi_2Se_3$ corresponds to a frequency of $\approx 40.9$ THz and $R_0 \approx 1.49$ nm. The dominant peak is associated to the transition $(n = -1, j = -1/2) \rightarrow (n = 1, j = 1/2)$. As expected, this peak shifts to lower frequencies for wires of larger radii.

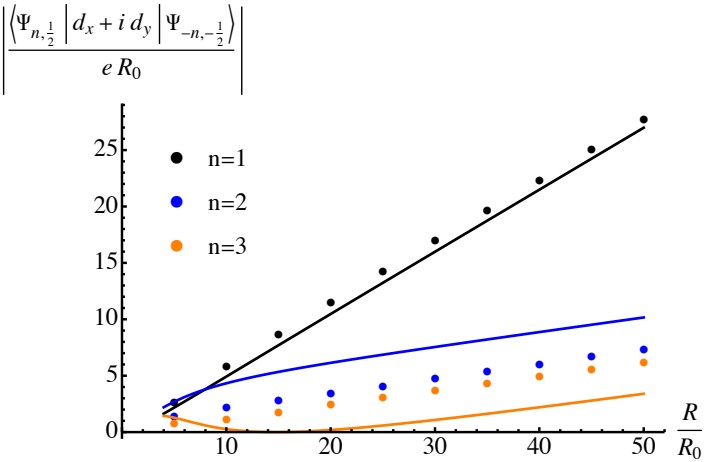

Figure 6: The magnitude of the matrix element of $d_+ = d_x + id_y$ for the transition $|\Psi_{-n,-1/2}\rangle \rightarrow |\Psi_{n,1/2}\rangle$ for $n = 1, 2$, and $3$ as a function of the radius and setting $L = 5R_0$. The points are obtained by means of the numerical tight-binding model with discretisation constants $a_r = 0.25R_0$ and $a_z = 0.05R_0$.

## 5   Conclusions

In this paper we theoretically investigate the topological properties of nanocylinders, with radius $R$ and height $L$, made of a 3D topological insulator. Making use of two models we focus on the calculation of the energy spectrum, the wavefunctions and the dipole matrix elements of the surface conducting states. Both models are based on an effective 2D Hamiltonian (with radial and longitudinal coordinates), which is obtained by making use of the cylindrical symmetry of the problem. The first model is an exact numerical tight-binding approach. The second one is an approximate analytical model which is based on a large-$R$ expansion of the lateral-surface states of the nanocylinder and accounts for its finite height through an effective boundary condition. In the following we list the main findings of the paper:

- By using the full (bulk) Hamiltonian for the wavefunctions of the lateral side of the cylinder we determine the quantisation rule for the momentum along the longitudinal direction, which is then used to derive analytical expressions for the eigenenergies and eigenfunctions. We find that such quantisation condition yields in general the correct dependence of the eigenenergies on the radius $R$ of the cylinder. In particular, in the case $j = 1/2$ the agreement is excellent.

- We calculate the detailed profile of the probability density for the four different components of the spinor wavefunction finding that, as expected, the surface state extends over the entire surface of the nanocylinder. Interestingly, the probability density on the two bases turns out to be small in the region around their centres. Moreover, we find that the four components of the wavefunction obtained analytically are a very good approximation of the ones obtained numerically, at least not too close to the bases.

- We calculate the dipole matrix elements for optical transitions with in-plane polarisation both numerically and analytically. These results are pertinent to current experimental

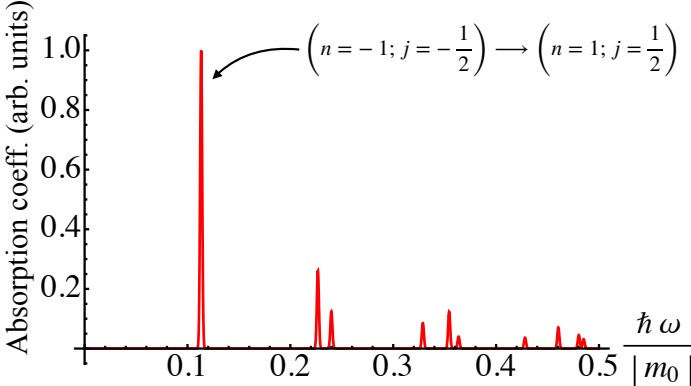

Figure 7: The absorption spectrum for a nanocylinder with $R = 20R_0$ and $L = 10R_0$. The peaks have been modelled by Gaussian curves with standard deviation $10^{-3}|m_0|$. The dominant peak corresponds to the transition $(n = -1, j = -1/2) \rightarrow (n = 1, j = 1/2)$. The dipole matrix elements have been computed by means of the tight-binding model with discretisation constants $a_r = 0.25R_0$ and $a_z = 0.05R_0$.

studies [62]. On the one hand, we derive the analytical expressions for such matrix elements when considering the contribution of the lateral side alone (which can be used to describe an infinite nanowire), or the bases alone, which can be used to model a squat nanocylinder. In the latter case, we find that the analytical results agree extremely well with the numerical exact results for the lowest energy transitions, while only describing their trend for higher energy transitions. On the other hand, we use the approximate model to derive analytical expressions for the matrix elements of a tall nanorod.

Our results can be used for the design of experiments exploring the surface states and the optical properties of finite-size nanocylinder.

# 6 Acknowledgements

We acknowledge Ulrich Zülicke for discussions on the theoretical results and Stephanie Law and Sivakumar Vishnuvardhan Mambakkam for discussions on the experimental realization and on the optical absorption of cylindrical QDs of TI. The computations were performed at the Rāpoi high performance computing facility of Victoria University of Wellington. MG wishes to thank Krista Steenbergen for help porting the code for the high-performance compute cluster.

# A    Eigenfunctions of the 2D approximated Hamiltonian $H_0$

We make the following Ansatz for the wave function of the effective 2D model, Eq. (7),

$$
\Phi_{k_z,j}(r,z) = \begin{pmatrix} a_1 \\ a_2 \\ a_3 \\ a_4 \end{pmatrix} \frac{e^{ik_z z}}{\sqrt{L}} e^{\lambda(r-R)}, \tag{40}
$$

where $k_z$ is the wavevector along the $z$-direction and $a_1, a_2, a_3$ and $a_4$ are numerical coefficients. Substituting the Ansatz in the Schrödinger equation for the Hamiltonian (11) we obtain the following system of equations for the coefficients

$$
\begin{pmatrix} m_0 - m_2\lambda^2 - E & 0 & 0 & -iA\lambda \\ 0 & -m_0 + m_2\lambda^2 - E & -iA\lambda & 0 \\ 0 & -iA\lambda & m_0 - m_2\lambda^2 - E & 0 \\ -iA\lambda & 0 & 0 & -m_0 + m_2\lambda^2 - E \end{pmatrix} \begin{pmatrix} a_1 \\ a_2 \\ a_3 \\ a_4 \end{pmatrix} = 0. \tag{41}
$$

Since we are looking for the surface states, we set the energy $E = 0$ and we find

$$
\lambda_\pm = \frac{A \pm \sqrt{A^2 + 4m_0 m_2}}{2m_2}. \tag{42}
$$

The solutions for the coefficients $(a_1, a_2, a_3, a_4)^T$ for $\lambda = \lambda_\pm$ are

$$
(1, 0, 0, i)^T \quad \text{and} \quad (0, 1, -i, 0)^T, \tag{43}
$$

where we have used the relation

$$
\frac{m_0 - m_2\lambda_\pm^2}{A\lambda_\pm} = -1. \tag{44}
$$

The eigenfunctions of the effective 2D Hamiltonian $H_0$ of Eq. (11) with $E = 0$ which fulfil the boundary conditions at $r = R$ are

$$
\Phi_{1,k_z,j}(r,z) = \frac{1}{\sqrt{2}} \begin{pmatrix} 1 \\ 0 \\ 0 \\ i \end{pmatrix} \frac{e^{ik_z z}}{\sqrt{L}} \rho(r) \tag{45}
$$

$$
\Phi_{2,k_z,j}(r,z) = \frac{1}{\sqrt{2}} \begin{pmatrix} 0 \\ 1 \\ -i \\ 0 \end{pmatrix} \frac{e^{ik_z z}}{\sqrt{L}} \rho(r), \tag{46}
$$

where the radial part of the wavefunction is defined as

$$
\rho(r) = \frac{1}{N} \left[ e^{\lambda_+(r-R)} - e^{\lambda_-(r-R)} \right], \tag{47}
$$

with $N$ being a normalisation factor.

# B   Matrix elements of the perturbation Hamiltonian $H_{1,j}$

In this appendix we diagonalise $H_{1,j}$ in the space spanned by $|\Phi_{1,k_z,j}\rangle$ and $|\Phi_{2,k_z,j}\rangle$, Eqs. (45) and (46), without need to worry about matrix elements between different values of $j$, since $j$ is conserved due to the cylindrical symmetry of the problem. The matrix elements of the perturbation $H_{1,j}$ are

$$\langle\Phi_{1,k_z',j}|H_{1,j}|\Phi_{1,k_z,j}\rangle = \delta_{k_z',k_z}\left(\frac{A}{R} - \frac{A^2}{2m_0}\frac{1}{R^2}\right)j \tag{48a}$$

$$\langle\Phi_{2,k_z',j}|H_{1,j}|\Phi_{2,k_z,j}\rangle = -\delta_{k_z',k_z}\left(\frac{A}{R} - \frac{A^2}{2m_0}\frac{1}{R^2}\right)j \tag{48b}$$

$$\langle\Phi_{2,k_z',j}|H_{1,j}|\Phi_{1,k_z,j}\rangle = \delta_{k_z',k_z}Bk_z \tag{48c}$$

$$\langle\Phi_{1,n',j}|H_{1,j}|\Phi_{2,n,j}\rangle = \delta_{k_z',k_z}Bk_z, \tag{48d}$$

where we have made use of the following approximations:

$$\langle\Phi_{l,k_z,j}|1/r|\Phi_{l,k_z,j}\rangle \approx \frac{1}{R} + \frac{m_2}{A}\frac{1}{R^2} - \frac{A}{2m_0}\frac{1}{R^2}$$

and

$$\langle\Phi_{l,k_z,j}|1/r^2|\Phi_{l,k_z,j}\rangle \approx 1/R^2$$

with $l = 1, 2$. As expected $k_z$ is also a good quantum number. The eigenergies are

$$E_{\pm,k_z,j} = \pm\sqrt{(Bk_z)^2 + j^2\frac{A^2}{R^2}\left(1 - \frac{1}{2}\frac{A}{m_0R}\right)^2}. \tag{49}$$

# C   Eigenfunction for the bases

We follow Ref. [65] in order to obtain the surface states of the top and bottom bases of the nanocylinder, but using polar coordinates to emphasize that the states are eigenstates of $\mathbf{J}_z$ due to the cylindrical symmetry. We assume that $L$ is large enough so that the effect of the hybridisation between the states localised at the top base and the bottom base is negligible. Therefore, we consider the two bases as independent. In the following, we will focus on the states localised around the top base, at $z = L$. We work with the original 3D Hamiltonian (1) and we choose for the eigenstate of the top base the following Ansatz in polar coordinates

$$\Psi_{\text{Top}}(r,\phi,z) = e^{\Gamma(z-L)}\frac{e^{ij\phi}}{\sqrt{2\pi}}\begin{pmatrix} c_1 J_{j-\frac{1}{2}}(k_\parallel r)e^{-i\frac{\phi}{2}} \\ c_2 J_{j-\frac{1}{2}}(k_\parallel r)e^{-i\frac{\phi}{2}} \\ c_3 J_{j+\frac{1}{2}}(k_\parallel r)e^{i\frac{\phi}{2}} \\ c_4 J_{j+\frac{1}{2}}(k_\parallel r)e^{i\frac{\phi}{2}} \end{pmatrix}, \tag{50}$$

where $J_n(z)$ is a Bessel function of the first kind and $k_\parallel$, $\Gamma$ and $c_1, c_2, c_3, c_4$ are numerical coefficients to be determined. Notice that we cannot make a large-$r$ expansion as the wave function is finite on the entire top surface of the cylinder and therefore also for small values of the radial coordinate.

Substituting the Ansatz in the Schrödinger equation with the Hamiltonian (1) we obtain the following relation between $\Gamma$, $k_\parallel$ and the energy

$$\left(k_\parallel^2 - \frac{m_1}{m_2}\Gamma^2 + \frac{m_0}{m_2}\right)^2 + \frac{A^2}{m_2^2}k_\parallel^2 - \frac{B^2}{m_2^2}\Gamma^2 - \frac{E^2}{m_2^2} = 0. \tag{51}$$

Solving for $\Gamma$ yields the two solutions with positive real part:

$$\Gamma_\pm(E) = \sqrt{\frac{B^2}{2m_1^2} + \frac{m_0}{m_1} + \frac{m_2}{m_1}k_\parallel^2 \pm \sqrt{\frac{B^4}{4m_1^4} + B^2\frac{m_0}{m_1^3} + \frac{E^2}{m_1^2} + \left(\frac{m_2B^2}{m_1^3} - \frac{A^2}{m_1^2}\right)k_\parallel^2}}. \tag{52}$$

There are four solutions for $(c_1, c_2, c_3, c_4)$, two for each $\Gamma = \Gamma_\pm$:

$$\left(i\frac{Ak_\parallel}{B\Gamma_\pm}, 0, -i, \frac{m_0 + m_2k_\parallel^2 - m_1\Gamma_\pm^2 - E}{B\Gamma_\pm}\right)^T \tag{53}$$

$$\left(i, \frac{m_0 + m_2k_\parallel^2 - m_1\Gamma_\pm^2 - E}{B\Gamma_\pm}, -i\frac{Ak_\parallel}{B\Gamma_\pm}, 0\right)^T. \tag{54}$$

The general solution for the wavefunction is a linear combination of the four independent solutions above. By imposing to that general solution the boundary condition $\Psi_{\text{Top}}(r, \phi, L) = 0$ we find the eigenenergies

$$E_\pm = \pm Ak_\parallel, \tag{55}$$

with $k_\parallel \geq 0$. The eigenfunctions corresponding to the energies $E_\pm = \pm Ak_\parallel$ are

$$\Psi_{\text{Top},+}(r, \phi, z) = \frac{e^{ij\phi}}{\sqrt{2\pi}}\begin{pmatrix} -iJ_{j-\frac{1}{2}}(k_\parallel r)e^{-i\frac{\phi}{2}} \\ J_{j-\frac{1}{2}}(k_\parallel r)e^{-i\frac{\phi}{2}} \\ iJ_{j+\frac{1}{2}}(k_\parallel r)e^{i\frac{\phi}{2}} \\ J_{j+\frac{1}{2}}(k_\parallel r)e^{i\frac{\phi}{2}} \end{pmatrix}\frac{1}{N_{\text{base},j}}\left[e^{\gamma_+(z-L)} - e^{\gamma_-(z-L)}\right] \tag{56a}$$

$$\Psi_{\text{Top},-}(r, \phi, z) = \frac{e^{ij\phi}}{\sqrt{2\pi}}\begin{pmatrix} iJ_{j-\frac{1}{2}}(k_\parallel r)e^{-i\frac{\phi}{2}} \\ -J_{j-\frac{1}{2}}(k_\parallel r)e^{-i\frac{\phi}{2}} \\ iJ_{j+\frac{1}{2}}(k_\parallel r)e^{i\frac{\phi}{2}} \\ J_{j+\frac{1}{2}}(k_\parallel r)e^{i\frac{\phi}{2}} \end{pmatrix}\frac{1}{N_{\text{base},j}}\left[e^{\gamma_+(z-L)} - e^{\gamma_-(z-L)}\right], \tag{56b}$$

where $N_{\text{base},j}$ is a normalisation constant which depends on $j$ and $k_\parallel$, but not on the sign of the energy in Eq. (55), and $\gamma_\pm = \Gamma_\pm(E = Ak_\parallel)$. Finally, we note the states of the top surface, Eqs. (56), can be also written in Cartesian coordinates as [65]

$$\Psi_{\text{Top},+,\mathbf{k}_\parallel}(\mathbf{r}_\parallel, z) = \frac{1}{2}\begin{pmatrix} ie^{-i\frac{\alpha}{2}} \\ -e^{-i\frac{\alpha}{2}} \\ -e^{i\frac{\alpha}{2}} \\ ie^{i\frac{\alpha}{2}} \end{pmatrix}\frac{e^{i\mathbf{k}_\parallel\cdot\mathbf{r}_\parallel}}{\sqrt{2\pi}}\frac{1}{N}\left[e^{\gamma_+(z-L)} - e^{\gamma_-(z-L)}\right] \tag{57a}$$

$$\Psi_{\text{Top},-,\mathbf{k}_\parallel}(\mathbf{r}_\parallel, z) = \frac{1}{2}\begin{pmatrix} -ie^{-i\frac{\alpha}{2}} \\ e^{-i\frac{\alpha}{2}} \\ -e^{i\frac{\alpha}{2}} \\ ie^{i\frac{\alpha}{2}} \end{pmatrix}\frac{e^{i\mathbf{k}_\parallel\cdot\mathbf{r}_\parallel}}{\sqrt{2\pi}}\frac{1}{N}\left[e^{\gamma_+(z-L)} - e^{\gamma_-(z-L)}\right], \tag{57b}$$

where $\mathbf{k}_\parallel = k_\parallel(\cos(\alpha), \sin(\alpha))$ and $\mathbf{r}_\parallel = (x, y)$. The states in Eqs. (57) agree with Ref. [65].

# D    Dipole matrix elements using the analytical model

When the contribution of the bases to the dipole matrix element are negligible, that is for the case of a tall nanorod $\gamma \ll 1$, we can use the approximate analytical state of Eq. (36) to obtain

$$\langle \Psi_{n,j'} | d_x \pm i d_y | \Psi_{-n,j} \rangle =$$

$$\frac{e^{-ik_\parallel(|n|,j')2R}}{2} e^{-ik_z(|n|,j)L} e^{-ik_z(|n|,j')L} \mathrm{sign}(j') \langle \Psi_{+,-k_z(|n|,j'),j'} | d_x \pm i d_y | \Psi_{-,k_z(|n|,j),j} \rangle \quad (58)$$

$$+ \frac{e^{i[k_\parallel(|n|,j)2R}}{2} e^{ik_z(|n|,j)L} e^{ik_z(|n|,j')L} \mathrm{sign}(j) \langle \Psi_{+,k_z(|n|,j'),j'} | d_x \pm i d_y | \Psi_{-,-k_z(|n|,j),j} \rangle$$

$$+ \frac{1}{2} e^{ik_z(|n|,j')L} e^{-ik_z(|n|,j)L} \langle \Psi_{+,k_z(|n|,j'),j'} | d_x \pm i d_y | \Psi_{-,k_z(|n|,j),j} \rangle$$

$$+ \mathrm{sign}(jj') \frac{e^{-ik_\parallel(|n|,j')2R} e^{ik_\parallel(|n|,j)2R}}{2} e^{ik_z(|n|,j)L} e^{-ik_z(|n|,j')L} \langle \Psi_{+,-k_z(|n|,j'),j'} | d_x \pm i d_y | \Psi_{-,-k_z(|n|,j),j} \rangle,$$

where $n > 0$. Now we calculate explicitly the absorption matrix element for $d_x + i d_y$ for circular polarisation for the transition $|\Psi_{-n,-1/2}\rangle \to |\Psi_{n,1/2}\rangle$. For $k_z > 0$ we find

$$\frac{\langle \Psi_{n,1/2} | d_x + i d_y | \Psi_{-n,-1/2} \rangle}{eR_0} = \frac{1}{2} \left( e^{-2ik_z L} e^{-ik_\parallel 2R} \langle \Psi_{+,-k_z,1/2} | d_x + i d_y | \Psi_{-,k_z,-1/2} \rangle - \quad (59) \right.$$

$$- e^{2ik_z L} e^{ik_\parallel 2R} \langle \Psi_{+,k_z,1/2} | d_x + i d_y | \Psi_{-,-k_z,-1/2} \rangle +$$

$$\langle \Psi_{+,k_z,1/2} | d_x + i d_y | \Psi_{-,k_z,-1/2} \rangle -$$

$$\left. \langle \Psi_{+,-k_z,1/2} | d_x + i d_y | \Psi_{-,-k_z,-1/2} \rangle \right).$$

Evaluating the matrix elements on the RHS by means of Eq. (37) we obtain the desired expression

$$\frac{\langle \Psi_{n,1/2} | d_x + i d_y | \Psi_{-n,-1/2} \rangle}{eR_0} = \left( \frac{R}{R_0} - 1 - \frac{m_2}{AR_0} \right) \times$$

$$\left[ -\frac{A}{2R} \frac{1}{E_{|n|,1/2}} + \frac{\sin(k_z(|n|,1/2)L)}{k_z(|n|,1/2)L} \cos(|n|\pi) \right].$$

We stress that this expression is valid when $\gamma \ll 1$. In this limit and in zeroth order in $\gamma$, this formula reduces to

$$\frac{\langle \Psi_{n,1/2} | d_x + i d_y | \Psi_{-n,-1/2} \rangle}{eR_0} = - \left( \frac{R}{R_0} - 1 - \frac{m_2}{AR_0} \right) \left[ 1 + \frac{\sin(1)}{|n|\pi - 1} \right].$$

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
