# Peer review of "Topological-Insulator Nanocylinders"

_SciPost Physics Core_

## Round 1 · Referee Report · Anonymous (Referee 1) · 2022-10-13

Strengths

In this work, the authors present a detailed calculation of the spectra and dipole matrix elements of a model for three dimensional topological insulator nanostructures with the form of a finite cylinder. Technically, the manuscript appears correct.

Weaknesses

The work is an extension to previous work (Ref. 54) where the same problem was considered for infinite cylinders by the same authors (among others). This work uses the exact same formalism of the continuum model, supplemented now by finite boundary conditions in the z direction. This same approach was followed by Kundu et al (which is not discussed in the paper)

https://journals.aps.org/prb/abstract/10.1103/PhysRevB.83.125429

In my opinion, the main problem with this paper is that it does not justify its purpose almost at all. The introduction describes topological insulator nanostructures very generally, and describes honestly the fact that infinite wires were worked out in Ref. 54 among many other references. The only motivation ever offered for the fact that now finite L is considered is “In this paper we focus on TI nanocylinders with finite L”. Why is the finite L case interesting? What does it offer compared to the previous paper? Currently, the paper reads as if the only motivation to do it is because it can be done. The conclusions similarly offer no physical insight, and reduce to describing to what extent numerical and analytical calculations agree. The paper would significantly improve if the authors explained what motivated them to carry out this calculation, what physical problem are they trying to solve, and what conclusions may one draw from their results that will motivate the study of such nanocylinders. In my opinion, these explanations in the introduction would be of much higher value to readers than connections to biomedical applications and cancer therapy.

In my opinion, the paper also falls short of providing any result that is directly relevant to the optics and THz community. The isolated calculation of the dipole matrix element is a very indirect way of approaching this problem, as opposed to providing an actual prediction of the optical conductivity or any other optical response. As provided in Fig. 6, what is the meaning of these results? Is this connected to any observable effect, and of what magnitude? What do we learn about the dependence of the matrix elements as a function of R?

Another pressing problem is that the manuscript does not describe the work of Kundu et al mentioned previously. This work from 2011 solves exactly the same problem and compares surface and bulk theories, compares with a true real space tight binding model, and in my opinion offers significantly more physical insight. The authors should compare their results with this paper and again explain what is the added value of their paper.

Report

In my opinion, the paper does not currently meet the criteria for SciPost Physics Core.

---

## Round 2 · Referee Report · Anonymous · 2023-1-18

Report

I thank the authors for their constructive response and the significant work in rewriting parts of the manuscript. I do believe the paper will be of more benefit to readers in the current form. The motivation to carry out the calculation is clear, the mentioned works are discussed appropriately, and the physical response associated to the dipole matrix elements is presented explicitly. I have no further comments and I believe the work is now suitable for SciPost Physics Core.

---

## Round 2 · Author Response

The Referee raises two points of criticisms, which we address and rebut in detail below.

The Referee's report has been of help to improve the manuscript and we are grateful for this. Following their comments we have extensively modified the introduction and conclusions sections and we have added a new figure (Fig. 7) in the manuscript.

Finally, we wish thank the Referee for bringing to our attention a relevant reference, which we have now included.

Detailed Answers

1) We agree with the Referee that our work is an extension of our previous work, although not a trivial one. The previous work dealt with infinitely-long cylinders. In the present manuscript we focus on finite-height cylinders, which are of current experimental interest (see reply to point 3). This is done with a two-pronged approach: a) we derive analytically a quantisation-condition for the longitudinal momentum; b) we use a finite-difference numerical model which is applicable to cylinders of any aspect ratio.

We wish to stress here that the quantisation condition that we find (which is fully validated by the comparison with the numerical results) differs from the one of Kundu et al.

We indeed overlooked the work mentioned by the Referee. However, our approach and the one of Kundu et al. do differ and as a consequence we find different results, as detailed in point 4.

2) We agree with the Referee that we did not elaborate sufficiently on the motivations for our work.
Obviously experimental realisations of topological-insulator nanocylinders are of finite height. And one possible way to characterise their topologically-protected surface states is by optical spectroscopy. Two important mechanisms affecting the optical absorption of a finite cylinder are neglected in the infinite-height limit considered in our previous paper Ref. 54. First, the finite height changes the energies of the surface states and therefore the frequencies of the absorption lines. Second, the wavefunction on the sides of the cylinder is affected by the finite height and moreover the bases also need to be accounted for. The fact that the wavefunction is different in the case of finite-height cylinders implies that the dipole matrix elements, which determine the strength of the absorption, are different than the one computed in the infinite-height limit.

Finally, there is the very interesting theoretical point of finding the correct quantisation condition for the longitudinal momentum entering the wavefunction of the lateral side surface of the cylinder. Indeed, the side surface state of the cylinder, when it approaches the top/bottom bases, bends and merges with the surface states on the bases and hard-wall boundary conditions cannot be imposed.

In the revised version of the paper we have emphasised these points.

3) We strongly disagree with the Referee on this point. The dipole matrix elements determine the absorption amplitudes in semiconducting low-dimensional structures [See H. Haug and S. W. Koch, Quantum Theory of the Optical and Electronic Properties of Semiconductors, 4th ed. (World Scientific, Singapore, 2004) and I. Pelant and J. Valenta, Luminescence Spectroscopy of Semiconductors (Oxford University Press, Oxford, UK, 2012).

In the case of transitions between surface states of topological-insulator cylinders of realistic radii the frequencies of the absorption lines are in the THz range.

We wish to mention that in the last months we have been providing data to the experimental group led by Stephanie Law who are measuring the absorption of THz radiation in nanocylinders made of Bi$_2$Se$_3$. Our calculations are very valuable in order to predict the frequencies and intensities of the absorptions lines.

The matrix elements plotted in Fig. 6 provide the absorption amplitude for THz radiation with circular polarisation. In particular, the dependence on the radius $R$ is an important prediction in order to maximise the optical response: the absorption amplitude is roughly linear with $R$ with a slope that depends on $n$.

In the revised version of the paper we have added a figure (Fig. 7) showing an example of the absorption spectrum.

4) We are grateful to the Referee for pointing this paper out, we had indeed overlooked the work of Kundu et al., which is definitely very relevant. However, we remark that the only overlap is the numerical solution for the eigenenergies and eigenfunctions. In the following we list the novel contributions of our paper:

a) The analytical approach in Kundu et al. was based on the Dirac-fermion (surface) theory, where only the linear terms are retained in the Hamiltonian. They have found as a quantisation condition for the longitudinal momentum the standard one, namely $k L=n\pi$. On the contrary, we use the full (bulk) Hamiltonian for the wavefunctions of the lateral side of the cylinder and propose a different quantisation condition. We find that such a quantisation condition yields the correct dependence of the eigenenergies on the radius $R$ of the cylinder. On the contrary, the standard quantisation condition of Kundu et al. does not predict the correct dependence on $R$ as can be easily shown by comparing Eq. (26) with $k_z=n\pi/L$ (the result of Kundu et al.) with our tight-binding results (Fig 1 of our manuscript) and our analytical results given by Eqs. (33) with $k_z$ given by Eq. (32). While we do not provide a figure with this comparison in the paper, we are happy to send it to the editor if required.

b) We calculate the probability density for the four different components of the spinor wavefunction, emphasizing peculiar behaviours close to the surfaces of the nanocylinder (for example the fact that ``the density on the two bases appears to be small in the region around their centres."). Moreover, we find that the four components of the wavefunction obtained analytically are a very good approximation for the ones obtained numerically, at least not too close to the bases.

c) We calculate the dipole matrix elements for optical transitions both numerically and analytically. These results are pertinent to current experimental studies. The optical properties of the cylinder are not addressed in Kundu et al. This is the added value of our paper, which we have now stressed in the revised version of the paper in the introduction, in the conclusions and in the results sections. On the contrary, the issue addressed by Kundu in their paper is mainly the spin and parity structure of the eigenstates. Their main finding is that the spin direction in a topologically protected surface mode is not locked to the surface.

---

## Round 2 · List of Changes

1) Extensive changes in the introduction;
2) Conclusions completely rewritten;
3) Small changes at the beginning of Section 4;
3) Added Fig. (7) and text describing it in Section 4;
4) Added 3 new References including to the paper by Kundu et al suggested by the Referee.

You are currently on this page

Resubmission 2209.00890v2 on 23 November 2022

---

## Editorial Decision

accepted_in_target_journal